# Safety of Global SARS-CoV-2 Vaccines, a Meta-Analysis

**DOI:** 10.3390/vaccines10040596

**Published:** 2022-04-12

**Authors:** Linyi Chen, Xianming Cai, Tianshuo Zhao, Bingfeng Han, Mingzhu Xie, Jiahao Cui, Jiayu Zhang, Chao Wang, Bei Liu, Qingbin Lu, Fuqiang Cui

**Affiliations:** 1Department of Epidemiology and Biostatistics, School of Public Health, Peking University, Beijing 100191, China; chenlinyi137@pku.edu.cn (L.C.); 1710306124@pku.edu.cn (X.C.); zts2018@pku.edu.cn (T.Z.); hanbingfeng@pku.edu.cn (B.H.); mingzhuxie@pku.edu.cn (M.X.); 2Faculty of Medicine, Imperial College London, London SW7 2AZ, UK; jiahao.cui22@imperial.ac.uk; 3Bioinformatics Program, School of Science, Xi’an Jiaotong-Liverpool University, Suzhou 215123, China; jiayu.zhang19@student.xjtlu.edu.cn; 4Department of Laboratorial Science and Technology & Vaccine Research Center, School of Public Health, Peking University, Beijing 100191, China; wchao@bjmu.edu.cn (C.W.); 1916387057@bjmu.edu.cn (B.L.); qingbinlu@bjmu.edu.cn (Q.L.)

**Keywords:** COVID-19, vaccine, safety, adverse events, meta-analysis

## Abstract

(1) Background: Severe acute respiratory syndrome coronavirus 2 (SARS-CoV-2) vaccines were developed in only a short amount of time and were widely distributed. We conducted this meta-analysis to understand the safety of SARS-CoV-2 vaccines. (2) Methods: We searched the corresponding literature published from 1 January 2020 to 20 October 2021. Information of adverse events (AEs) of each selected work was collected. The quality and bias of studies was evaluated, and meta-analysis was carried out by using Stata 17.0. (3) Results: Totally, 11,451 articles were retrieved, and 53 of them were included for analysis. The incidence rate of AEs was 20.05–94.48%. The incidence rate of vascular events increased after viral vector vaccination, while the incidence rate of vascular events decreased after mRNA vaccination. Viral vector vaccine had a higher AE rate compared to mRNA vaccines and inactivated vaccines. In most circumstances, the incidence of AEs was higher in older people, female and after the second dose. The sensitivity of meta-analysis was acceptable; however, the literature was subject to a certain publication bias. (4) Conclusions: The safety of SARS-CoV-2 vaccines was acceptable. The incidence of allergic symptoms and cardiovascular and cerebrovascular symptoms was low. Viral vector vaccine had a higher risk of leading to thrombosis events. The understanding of SARS-CoV-2 vaccine AEs should be enhanced, so as to promote the vaccination.

## 1. Background

Coronavirus disease 2019 (COVID-19) emerged in December 2019 and has since spread around the world [1], causing more than 450 million infections and 6 million deaths worldwide up to March 2022 [2]. On 30 January 2020, the World Health Organization (WHO) declared COVID-19 as a public health emergency of international concern (PHEIC) [3]. COVID-19 is highly contagious, and no specific drug has been developed, thus vaccination became an important measure to control the outbreak. Up to 25 January 2022, a total of 10 SARS-CoV-2 vaccines have been included in the WHO emergency use list (EUL), and 33 SARS-CoV-2 vaccines have been approved for use in at least one country [4]. The safety of SARS-CoV-2 vaccines is of great concern due to the short development period and wide range of vaccination. At present, there are few comprehensive analyses of SARS-CoV-2 vaccines safety, especially systematic meta-analyses. Therefore, we conducted a meta-analysis to assess the safety of SARS-CoV-2 vaccines.

Currently, there are five main development approved paths for SARS-CoV-2 vaccines worldwide, which are mRNA vaccine, DNA vaccine, non-replicating viral vector vaccine, inactivated vaccine and protein subunit vaccine. DNA vaccines uses plasmids to transmit virus DNA into human cells to produce antigen proteins. However, if the plasmid DNA were to integrate into the human genome, tumors may occur [5]. mRNA vaccine is an emerging platform. Its technique is transporting the mRNA that encodes virus antigen into host cell through lipid nanoparticles. Inactivated vaccine is the most mature virus vaccine development technology, which was used in the hepatitis A, hand-foot-and-mouth disease, and influenza vaccines as well as other products. By killing a cultured virus by physical or chemical methods, an inactivated vaccine was made [6]. Viral vector vaccine uses weak or non-pathogenic viruses as a vector and integrates antigen genes of target virus into the genome of the vector. The recombinant virus can express the antigenic protein of the target virus in vivo and induces an immune response [5]. For the differences of characteristics of vaccine types, we carried out the meta-analysis based on vaccine types.

Most vaccines required two doses, intramuscular, and were 21 to 28 days apart. Individual vaccines require one or three doses of injection, such as CanSino vaccine (one dose) and Anhui Zhifei vaccine (three doses). Different vaccines were approved in different countries. For example, the Ad26.COV2.S vaccine was widely approved in North and South America and Europe, while CoronaVac was more widely used in Asia.

There are differences between adverse effect and adverse events. While an adverse effect is an adverse outcome that can be attributed to some action of a drug [7], an adverse event (AE) is any undesirable experience associated with the use of a medical product in a patient that does not necessarily have a causal relationship with the treatment [8,9]. Vaccine adverse events can be divided into general events and abnormal events: general events are transient events caused by the inherent characteristics of the vaccine, such as local redness, pain, general discomfort, tiredness and other comprehensive symptoms; abnormal events include more severe systemic events and allergic events [10]. For the rare but acute morbidity and serious prognosis, this study focused on abnormal events including thrombosis, anaphylaxis and other events. The incidence rates of AEs were different on different age groups, gender, study location and underlying diseases of the population. In this study, we collected the incidence of AEs, basic information and vaccine development platform of phase III, phase IV RCT and the real-world study. Subgroup analysis was conducted according to the research type, vaccine development platform, age and gender of the study population.

## 2. Materials and Methods

### 2.1. Literature Retrieval

We searched for bibliography from Pubmed, Embase, Cochrane COVID-19 Study Register, Cochrane Library, Chinese National Knowledge Infrastructure (CNKI), Wanfang Data Knowledge Service Platform (Wanfang) and SinoMed. The searching strategy was to search keywords, titles or abstracts containing ‘COVID-19 vaccines’, ‘safety’, ‘AEs’, ‘tolerance’, ‘RCT’, ‘clinical trial’, ‘real world study’ and the corresponding Chinese words and were published during 1 January 2020 to 20 October 2021. Then, we screened the bibliography according to the including and excluding criteria.

### 2.2. Literature Screening

Including criteria: (1) study subjects were people who had been vaccinated of SARS-CoV-2 vaccine, including special population; (2) the study reported the number and the incidence rate of specific AEs, or the rate could be calculated according to the data; (3) experimental and observational studies, such as RCT on phase III and IV, cross-sectional study, cohort study, surveillance data, etc.; and (4) language was Chinese or English. Excluding criteria: (1) repeated publications; (2) only the research plan instead of result was reported; (3) reviews, comments, lectures, or news reports; and (4) no documentation of the development platform of the vaccine.

### 2.3. Data Extraction and Quality Assessment

We used EndNote X9 (Bld 12062) to conduct literature management. After importing the literature, duplicate works were removed by computer retrieval. Then, the screening was carried out by reading the title and the abstract. Literature that met the criteria was included after reading the full text. A table was established to extract the following contents from the literature: title, author, publication year, study type, study site, vaccine name, vaccine development platform, study population, total number of vaccinated people, number and the incidence of each AE, main conclusions and study limitations. Literature retrieval and information extraction were carried out by two people (Linyi Chen and Xianming Cai) in parallel. In case of inconsistency, consensus was reached through discussion or consultation with experts.

In the cross-sectional study, the 11-item scale recommended by Agency for Healthcare Research and Quality (AHRQ) [11] was used to evaluate the literature quality. One point was scored if the item was in conformity, and zero was scored if the item was not in conformity or unclear. The total score 1–3 was low quality, 4–7 was medium quality, and 8–11 was high quality. The quality of cohort studies was evaluated using The Newcastle-Ottawa Scale (NOS) [12], and the quality of RCTs was evaluated using the Cochrane collaboration’s tool for assessing risk of bias.

In this paper, we only focus on abnormal events, including allergic symptoms, cardiovascular and cerebrovascular symptoms.

### 2.4. Statistical Analysis

Meta-analysis was conducted according to different development platforms, which were inactivated vaccines that included CoronaVac and BIBP/WIBP, mRNA vaccines that included BNT162b2 and mRNA-1273, and viral vector vaccines that included Ad26.COV2.S, ChAdOx1 and Sputnik V. If the number of events was zero, Bartlett correction was used, which was adding one to the number of events and multiplying the total number of vaccinated people by four. In this study, Stata 17.0 (Stata Corp LP, College Station, TX, USA) was used to calculate the pooled rate and 95% confidence interval (95%CI) of AEs, severe AEs, allergic symptoms, cardiovascular and cerebrovascular symptoms after COVID-19 vaccination. As it was not likely that intervention effects across studies are identical, we chose random-effects model for all AEs meta-analysis [13]. “Leave-one-out” method was used for sensitivity analysis. Pooled rate was calculated when excluding one study at each analysis to see if the result changed significantly. Result was considered as robust if the pooled rates were located inside the 95%CI of original pooled rate.

## 3. Results

### 3.1. Literature Retrieval Results

We retrieved 11,486 works published from 1 January 2020 to 20 October 2021 in total. Among them, 3028 were from PubMed, 381 were from Embase, 6231 were from Cochrane COVID-19 Study Register, 730 were from Cochrane Library, 684 were from CNKI, 337 were from Wanfang and 95 were from SinoMed. After removing duplicates, there were 5494 left. After screening the literature by reading the topics and abstracts, there were 375 left. After reading the full texts, the 53 left were included into this study (Figure 1).

### 3.2. Literature Quality Evaluation

Of the 53 included articles, 38 were cross-sectional studies, nine were cohort studies, and six were RCTs. Different quality evaluation scales were used for different types of studies. There were 11 items in cross-sectional studies evaluation, and each item scored one point. The quality score of cross-sectional studies was between four and ten, which were medium to high quality (Table A1). Most studies met the requirements of defining the source of information, indicating time period and clarifying what follow-up was expected. The main items that were not satisfied were ‘Indicate if evaluators of subjective components of study were masked to other aspects of the status of the participants’, ‘Explain any patient exclusions from analysis’ and ‘Describe how confounding was assessed and/or controlled’. There were eight items in cohort studies evaluation. The score of the fifth item (compare ability of cohorts) was two points, and the score of other items were one point. Cohort study quality scores ranged from six to nine, which were medium to high quality (Table A2). All studies met the requirement of demonstrating that outcome of interest was not present at the start of study. Most studies met the requirements of selecting a non-exposed cohort and ascertaining exposure. The main item that was not satisfied was ‘Compare ability of cohorts on the basis of the design or analysis’. There were seven items in RCT evaluation. All RCTs met the requirements of blinding of participants and personnel, and selective reporting. Study 11 and 30 did not clarify the method used to generate the allocation sequence. Study 30 did not describe the method used to conceal the allocation. Study 4 did not describe the completeness of outcome data. Five RCTs did not mention how outcome assessors were blinded from knowledge of which intervention a participant received (Figure 2 and Table A3). Overall, risk of bias for RCTs was low. As the overall quality of research literatures were medium to high, and risk of bias was low, study quality was not the main reason causing high heterogeneity of results.

### 3.3. Basic Information

The 53 articles included were all published in 2021, and the study sites included the United States (34%), the United Kingdom (9%), Korea (9%) and other countries, and the vaccines included Ad26.COV2.S, BNT162b2, mRNA-1273, ChAdOx1, Sputnik V, CoronaVac, and BIBP/WIBP. The control group received normal saline, aluminum adjuvant or other placebos. People origin included the general population, healthcare workers (HCW), organ transplant recipients, patients with specific diseases, etc., and the number of participants in each group ranged from 26 to 265 million (Table 1). When there was no overall result of the AEs in the literature, the results of the first dose were included in the analysis as a representative.

### 3.4. Safety Analysis of SARS-CoV-2 Vaccines

#### 3.4.1. Overall Situation

Meta-analysis of AEs was conducted according to the three vaccine development platforms, respectively. The I^2^ of most AEs was above 90%, which indicated the incidences were with great heterogeneity among studies. The high heterogeneity might be due to the differences of population origins, sample size and sensitivity. Meta-analysis showed that in observational study, the incidence of general AEs of viral vector vaccine, mRNA vaccine and inactivated vaccine ranged from 20.05% to 94.23% (Figure 3), and the incidence of serious AEs (SAEs) ranged from 0.07% to 1.25%. The incidence of death was 0.03% doses, which was only reported after mRNA vaccines inoculation. Totally, four studies (six records in total, for two studies have records of both exposed group and control group) investigated the incidence of death of mRNA vaccine receivers, three deaths were reported among 137 patients with cancer. No death was reported in the other three studies. For allergic symptoms, the incidence rates of tongue edema, angioedema, body bruises, urticaria, and anaphylaxis were 3.61–18.04%, 2.24–9.58%, 1.98–4.96%, 0.91–2.99%, and 0.24–0.95%, respectively. For cardiovascular and cerebrovascular symptoms, the incidence rates of palpitation, irregular heartbeat, abnormal blood pressure and chest discomfort were 0.09–28.29%, 7.81–23.03%, 2.47–13.85% and 1.68–13.21% (Table 2).

In surveillance studies, the incidence of general AEs, SAEs and death of viral vector vaccine and mRNA vaccine was 197.0/100,000 doses and 131.7/100,000 doses, 35.1/100,000 doses and 3.6/100,000 doses, and 0.44/100,000 and 0.35/100,000 doses, respectively. For allergic symptoms, the incidence of body bruises was 0.9–2.4%. For cardiovascular and cerebrovascular symptoms, the incidence of venous thrombosis, arterial thrombosis, heart events, thrombocytopenia, and cerebral venous sinus thrombosis (CVST) were 34.0/100,000 doses and 31.2/100,000 doses, 125.8/100,000 doses and 147.3/100,000 doses, 35.3/100,000 doses and 14.0/100,000 doses, 4.6/100,000 doses and 7.4/100,000 doses, and 0.32/100,000 doses and 2.27/100,000 doses for viral vector vaccines and mRNA vaccines, respectively.

In RCTs, the incidence of general AEs was 64.8%, 57.3% and 43.0% of viral vector vaccine, mRNA vaccine and inactivated vaccine, respectively. The incidence of SAEs was 0.5%, 1.1%, and 0.5%, respectively. The incidence of death was 0.02% and 0.01% for viral vector vaccine and mRNA vaccine. Only studies of viral vector vaccine reported allergic symptoms and cardiovascular and cerebrovascular symptoms. The incidence of urticaria and anaphylactoid was 36.4/100,000 doses and 41.1/100,000 doses. The incidence of CVST, venous thrombosis, pulmonary embolism and pericarditis was 4.57/100,000 doses, 50.2/100,000 doses, 18.27/100,000 doses and 4.57/100,000 doses for viral vector vaccines.

#### 3.4.2. Comparison with the Unvaccinated Population

Compared with the unvaccinated population or population that received placebo, surveillance study showed that people inoculated with viral vector vaccine had a higher risk of having coagulation disorder/purpura (RR = 1.960, 95%CI 1.252–3.068), arterial thrombosis (RR = 1.167, 95%CI 1.103–1.234) and venous thrombosis (RR = 1.128, 95%CI 1.023–1.244). People inoculated with mRNA vaccine had a lower risk of having venous thrombosis (RR = 0.735, 95%CI 0.622–0.867) and arterial thrombosis (RR = 0.763, 95%CI 0.648–0.899) than the unvaccinated people. There were no data available for calculating RR after inactivated vaccine inoculation.

Results of RCT showed people inoculated with viral vector vaccine had lower incidence of death (RR = 0.305, 95%CI 0.124–0.745) and SAEs (RR = 0.815, 95%CI 0.681–0.976), higher incidence of venous thrombosis (RR = 3.665, 95%CI 1.023–13.137) than unvaccinated people. After receiving the mRNA vaccine, people had higher incidence of general AEs (RR = 1.998, 95%CI 1.653–2.414) and SAEs (RR = 1.727, 95%CI 1.403–2.127) than unvaccinated people. After receiving inactivated vaccine, people had lower incidence of general AEs (RR = 0.925, 95%CI 0.875–0.978) and SAEs (RR = 0.747, 95%CI 0.590–0.945) than the unvaccinated people.

#### 3.4.3. Comparison among Different Types of Vaccines

This paper included three vaccine development routes: viral vector vaccine, mRNA vaccine and inactivated vaccine. In observational studies, the incidence of general AEs, palpitation, irregular heartbeats, tongue edema, abnormal blood pressure, chest discomfort, angioedema, body bruises, urticaria and anaphylactoid of mRNA vaccines or inactivated vaccines were lower than those of viral vector vaccine. SAEs were with a higher incidence rate when inoculated with mRNA vaccines than those of the other two types of vaccines (Figure 4A).

In surveillance studies, there was only data of viral vector vaccine and mRNA vaccine. The incidence of general AEs, SAEs, death, body bruises, venous thrombosis, coagulation disorder/purpura, heart events was higher after viral vector vaccine inoculation than those of mRNA vaccine. The incidence of CVST, thrombocytopenia and arterial thrombosis were higher after the injection of mRNA vaccine than those of viral vector vaccine.

In RCTs, the incidence of general AEs, death, was highest after viral vector vaccine inoculation. The incidence of SAEs was the highest after mRNA vaccine inoculation. The data of allergic symptoms and cardiovascular and cerebrovascular symptoms were not available for mRNA vaccines and inactivated vaccines.

#### 3.4.4. Comparison between Different Doses

All vaccines included in this meta-analysis required two doses. In observational studies, the incidence of most AEs including general AEs, fast heartbeat, urticaria, abnormal blood pressure, bad rash all over the body, chest discomfort, anaphylaxis, pericarditis and dead, was higher after the second dose of mRNA vaccine than those after the first dose. Palpitation happened more frequently after the first dose of mRNA vaccine than those after the second dose. For inactivated vaccines, the incidence of general AEs was higher after the first dose than after the second dose (Figure 4B).

In surveillance studies, after mRNA vaccine inoculation, the incidence of body bruises was higher after the first dose than after the second dose. In RCTs, after mRNA vaccine inoculation, the incidence of general AEs was higher after the second dose than after the first dose. Data was not complete to compare the incidence between doses of other AEs.

#### 3.4.5. Comparison between Different Age Groups

In observational studies, the incidence of general AEs, palpitation, angioedema, tongue edema, hypotension, chest discomfort and urticaria after viral vector vaccine were higher in people <55 years old than that in people ≥55 years old. For mRNA vaccine, the incidence of general AEs, palpitation, angioedema, tongue edema, and hypotension was higher in people ≥55 years old, while the incidence of chest discomfort and urticaria was higher in people <55 years old (Figure 4C).

In surveillance studies, after viral vector vaccine or mRNA vaccine inoculation, the incidence of thrombocytopenia, venous thrombosis, arterial thrombosis, and coagulation disorder/purpura in people ≥55 years old was higher than those in people <55 years old, except the incidence of coagulation disorder/purpura after mRNA vaccines inoculation was higher in people <55 years old than in people ≥55 years old. RCT data were insufficient to carry out comparison between age groups for each vaccine platform.

#### 3.4.6. Comparison between Different Genders

There were two studies that report the incidence of AEs of interest in different genders. The reported vaccine included ChAdOx1 and BNT162b2. Meta-analysis results showed that in observational studies, the incidence of general AEs, palpitation, angioedema, hypertension, hypotension, tongue edema, chest discomfort and urticaria was higher in females than in males after viral vector vaccine. After inoculation of mRNA vaccine, the incidence of general AEs, chest discomfort and urticaria was higher in females than in males, and the incidence of palpitation, angioedema, hypertension, hypotension and tongue edema was higher in males than in females (Figure 4D). Surveillance and RCT data were not sufficient to compare the incidence of AEs between genders.

### 3.5. Sensitivity and Publication Bias

In surveillance studies, the incidence of SAEs of viral vector vaccine increased when omitting study “43” (Table A4). The incidence of anaphylaxis of mRNA vaccine decreased when omitting the data of mRNA-1273 vaccine of study “44”, and increased when omitting the data of BNT162b2 vaccine of study “44” (Table A5). No outliers were detected in observational studies or RCTs.

The Egger’s test was used to evaluate the publication bias of literature. When *p* < 0.05, publication bias was considered. In observational studies, the incidence of death (*p* = 0.001), anaphylaxis (*p* < 0.001) and chest discomfort (*p* = 0.032) of mRNA vaccine showed publication biases. No publication bias was detected for viral vector vaccines and inactivated vaccine inoculation.

Funnel plot of incidence rates of AEs showed publication bias in observational studies. Studies of mRNA vaccines had overestimated the incidence of AEs (Figure A1). The number of surveillance studies and RCT studies was too small to show publication bias.

## 4. Discussion

A total of 53 works were included in this study, covering a wide range of regions, diverse population sources and research types, which can provide a comprehensive summary of current AEs of COVID-19 vaccination. The overall quality of the data was high, and the results of those studies were reliable.

The WHO has included ten SARS-CoV-2 vaccines in its EUL, which have passed the evaluation of quality, safety, efficacy and other indicators [67]. WHO’s safety requirements for SARS-CoV-2 vaccines were that the benefits of vaccines largely outweigh their safety risks [68]. SARS-CoV-2 vaccines were with effective protection against the infection [69]. Studies showed mRNA vaccine was with highest efficacy (94.5–95%), followed by viral vector vaccines (66.7–91.6%) and inactivated vaccines (72.8–83.5%) [69]. Our study provided evidence to support promoting vaccine campaigns in the COVID-19 pandemic. In this study, we found that the safety of SARS-CoV-2 vaccines was acceptable, that most AEs were mild and transient, and the incidence of SAEs was low. The overall incidence of AEs of SARS-CoV-2 viral vector vaccines, mRNA vaccines and inactivated vaccines of observational studies, surveillance studies and RCTs were at medium to high level (20.05–94.48%). The incidence of SAEs was low (3.56/100,000–1.25%). The outcome of SAEs can be improved by increasing attention, strengthening the surveillance and timely treatment. Allergic symptoms and cardiovascular and cerebrovascular symptoms were with low incidence rates (0.32/100,000 doses–28.3%). Anaphylaxis is a severe, life-threatening systemic hypersensitivity reaction characterized by rapid onset with airway, breathing, or circulatory problems. Individuals who develop anaphylaxis to a SARS-CoV-2 vaccine should not receive a second dose of that vaccine [70]. However, for patients with urticaria or angioedema after SARS-CoV-2 vaccine inoculation, no consensus of whether they should receive a booster is reached. Once anaphylaxis is detected, intramuscular epinephrine should be started immediately [71].

Compared with unvaccinated group or the placebo group, vaccination of viral vector vaccine may increase the risk of thrombosis, but decrease the risk of death. The incidence of vascular diseases decreased after receiving the mRNA vaccine. These results revealed the relationship between thrombosis events and viral vector vaccines. The incidence of AEs after inoculation by inactivated vaccines was lower than control groups that showed good safety of inactivated vaccines. Vaccine-related thrombotic events are caused by vaccine-induced immune thrombotic thrombocytopenia (VITT). VITT is a rare but life-threatening AE. First death caused by SARS-CoV-2 vaccine-related thrombosis was reported in February 2021. VITT was mainly reported 5–24 days after viral vector vaccine inoculation [72]. VITT could cause thrombosis in human body parts, especially CVST and visceral vein thrombosis. The mechanism of VITT was not clear. Effect of anti-PF4 antibodies [73], adenovirus vector [74] or S protein of SARS-CoV-2 [75] could have led to VITT. For patients with VITT, intravenous immunoglobulin should be started immediately to lower the risk of thrombosis events [76]. Due to its serious consequences, surveillance of VITT should be strengthened, corresponding education to the public should be reinforced in order to detect and treat in time.

Our study showed viral vector vaccine has a higher incidence of AEs than mRNA vaccine and inactivated vaccine. Viral vector vaccines use live viruses as vectors and can induce a strong immune response. Due to the existence of preexisting immunity of the vector viral, efficacy of a viral vector vaccine may be reduced. Inactivated vaccine is a mature development platform with good immunogenicity and safety, but is less immunogenic, often requiring adjuvants and repeated injections [6]. mRNA vaccine is a novel vaccine type with good safety and a simple production process that is suitable for an emergency vaccine [77].

The incidence of most AEs after the second dose of mRNA vaccine was higher than that after the first dose, and the incidence of AEs after the second dose of inactivated vaccine was lower than that after the first dose. SARS-CoV-2 vaccines generated higher peaks of anti-spike IgG after the second dose than after the first dose [78]. However, there is no specific correlation between incidence of adverse events and immunogenicity.

The incidence of most AEs was higher in the population <55 years old than in the population ≥55 years old. The mechanism of this phenomenon may be that the immune system of the elderly is not as strong as that of the young population and the immune response induced by the vaccine is weaker [46]. T cells and B cells become dysfunctional with age, that leads to lower induction of functional vaccine-specific antibodies [79].

The incidence of most allergic and cardiovascular and cerebrovascular AEs of viral vector vaccines was higher in females than in males, which may be due to the different immune responses of different genders to antigens [80]. Research revealed that females tend to show greater antibody responses, higher basal immunoglobulin levels and higher B cell numbers than males. The difference of sex chromosomes, hormonal mediators and environmental mediators between genders may also contribute to the difference level of immune responses [81]. Our result showed the incidence of AEs was higher in males than females after mRNA vaccine inoculation, but the mechanism of this phenomenon is unclear. Studies showed males have a higher incidence rate of serious AEs than females [80].

This study does have, however, some limitations. First, the comparability of data is affected by different definitions of symptoms, reporting methods, reporting personnel and observation time. Due to those factors, the heterogeneity was high. Therefore, the results should be interpreted cautiously. As subgroup analysis was conducted according to the studies types, vaccine types, number of doses, ages and gender, and no significant I^2^ decreasing was found, there existed other factors that contribute greatly to the high heterogeneity. In studies with a large study population, the incidence rate is lower than that in study with small population. This may be due to the difficulty in collecting minor AEs in larger reporting system. Another meta-analysis of SARS-CoV-2 vaccine safety also supported this phenomenon [82]. We employed strict including criteria, quality assessment and subgroup analysis to control heterogeneity. Second, the study population was diverse, including general population, HCW, cancer patients, people with history of severe allergies which is highly heterogeneous and has a certain selection bias. HCWs may be more sensitive to AEs therefore induced selection bias and overestimated the incidence. A meta-analysis of SARS-CoV-2 vaccine safety showed that the reported rate of AEs in HCW and patients was much higher than in the general population [82]. Third, AEs includes all symptoms and diagnoses happened after vaccination and do not necessarily have causal relationship with the vaccine itself. AEs could be anxiety-related reactions or coincidental events [83]. Therefore, this result could not be interpreted as the level of adverse reactions due to SARS-CoV-2 vaccine intrinsic quality, except the result of comparison between vaccinated and unvaccinated people.

WHO has been advocating a global COVID-19 vaccination strategy, aiming to reach the target of 70% of the population vaccinated by mid-2022. To achieve this goal, all countries must make further efforts [84]. Therefore, knowledge of SARS-CoV-2 vaccine needs to be strengthened to better promote vaccination. Further research may compare the incidence of AEs of SARS-CoV-2 vaccines with other vaccines in order to evaluate the safety of SARS-CoV-2 vaccines better. At present, more countries carry out the third or even the fourth dose of booster vaccination, and the results of booster shots need to be evaluated continuously. In addition, as the virus type evolves, health and economic evaluation should be carried out persistently to provide recommendations for COVID-19 vaccination strategies.

## 5. Conclusions

This meta-analysis showed that the safety of SARS-CoV-2 vaccines was acceptable. The incidence of allergic symptoms, and cardiovascular and cerebrovascular symptoms, was low. Viral vector vaccine had higher risk of leading to thrombosis events. In most circumstances, the incidence of AEs was higher in younger people, female and after the second dose. Our result supports promoting vaccine campaigns in the COVID-19 pandemic. Surveillance of anaphylaxis and thrombosis events after vaccination should be strengthened in order to provide treatment in time.

## Figures and Tables

**Figure 1 vaccines-10-00596-f001:**
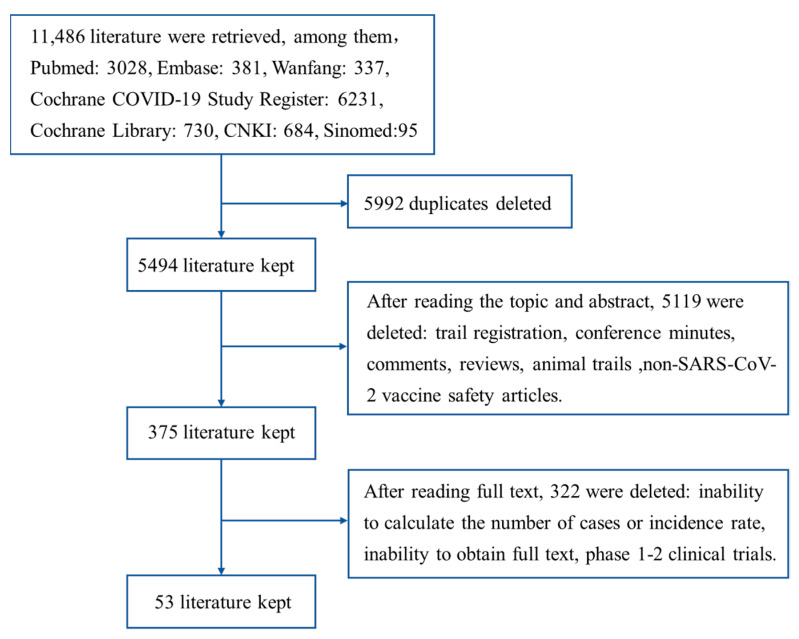
Literature screening process.

**Figure 2 vaccines-10-00596-f002:**
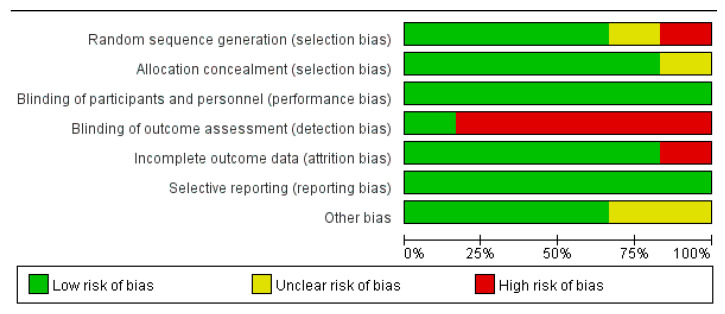
Risk of bias graph.

**Figure 3 vaccines-10-00596-f003:**
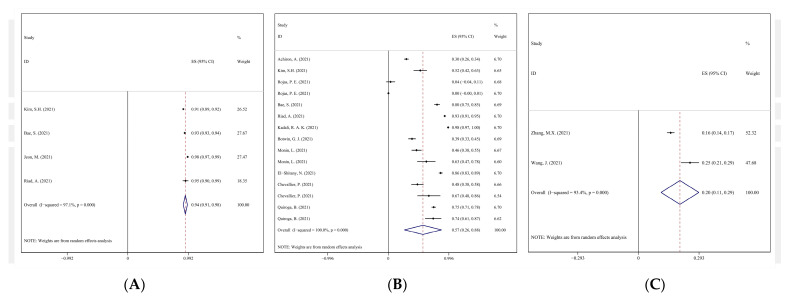
Forest plots of AEs in observational studies. (**A**) Forest plot of viral vector vaccine; (**B**) forest plot of AEs of mRNA vaccine in observational study; (**C**) forest plot of inactivated vaccine in observational study.

**Figure 4 vaccines-10-00596-f004:**
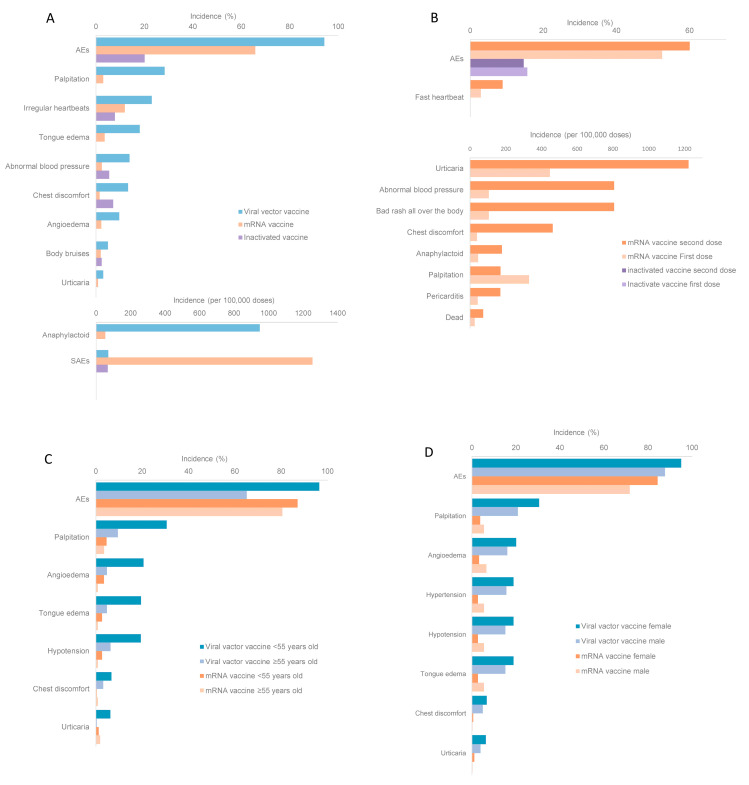
The incidence of adverse events (AEs) of different vaccines and population. (**A**) The incidence of AEs of different vaccines; (**B**) the incidence of dose-specific AEs with mRNA vaccine or inactivated vaccine inoculation; (**C**) the incidence of age-specific AEs with viral vector vaccine or mRNA vaccine inoculation; and (**D**) the incidence of gender-specific AEs with inactivated vaccine inoculation.

**Table 1 vaccines-10-00596-t001:** Basic information of the included literature.

ID	Author	Study Type	Study Site	Vaccination	People Origin	Population
1	Shay et al. [14]	Surveillance	the USA	Ad26.COV2.S	General population	338,765
2	Boyarsky et al. [15]	Cross-sectional study	the USA	BNT162b2 or mRNA-1273	Solid Organ Transplant Recipients	187
3	Menni et al. [16]	Surveillance	the UK	BNT162b2	General population	282,103
				ChAdOx1	General population	345,280
4	Sadoff et al. [17]	RCT	Argentina, Brazil, Chile, etc.	Ad26.COV2.S	General population	3356
5	Song et al. [18]	Cross-sectional study	Korea	ChAdOx1	HCW	2426
				BNT162b2	HCW	52
6	Achiron et al. [19]	Cohort Study	Israel	BNT162b2	Patients with multiple sclerosis	555
7	Kim et al. [20]	Cross-sectional study	Korea	ChAdOx1	HCW	1431
				BNT162b2	HCW	80
8	Waissengrin et al. [21]	Cohort Study	Israel	BNT162b2	Patients with cancer	137
9	Rojas et al. [22]	Cross-sectional study	Spain	BNT162b2	HCW with previous severe allergic diseases	26
				mRNA-1273	HCW with previous severe allergic diseases	104
10	Gee et al. [23]	Surveillance	the USA	BNT162b2 or mRNA-1273	General population	1,602,065
11	Voysey et al. [24]	RCT	the UK, Brazil, South Africa	ChAdOx1	General population	12,282
12	Logunov et al. [25]	RCT	Russia	Sputnik V	General population	16,427
13	CDC COVID-19 Response Team [26]	Surveillance	the USA	mRNA-1273	General population	4,041,396
14	CDC COVID-19 Response Team [27]	Surveillance	the USA	BNT162b2	General population	1,893,360
15	Bae et al. [28]	Cross-sectional study	Korea	ChAdOx1	HCW	5589
				BNT162b2	HCW	277
16	Zhang et al. [29]	Cross-sectional study	China	CoronaVac	HCW	1526
17	Wang et al. [30]	Cross-sectional study	China	BIBP/WIBP	HCW	4458
18	Kadali et al. [31]	Cross-sectional study	the USA	BNT162b2	HCW	803
19	Riad et al. [32]	Cross-sectional study	Czech Republic	BNT162b2	HCW	877
20	Al Kaabi et al. [33]	RCT	United Arab Emirates	WIBP	General population	13,478
				BIBP	General population	13,465
21	Kadali et al. [34]	Cross-sectional study	the USA	mRNA-1273	HCW	432
22	Jeon et al. [35]	Cross-sectional study	Korea	ChAdOx1	HCW	994
23	Wang et al. [36]	Cross-sectional study	China	BIBP/WIBP	patients with non-alcoholic fatty liver disease	381
24	Ou et al. [37]	Cohort Study	the USA	BNT162b2 or mRNA-1273	Solid Organ Transplant Recipients	741
25	Botwin et al. [38]	Cross-sectional study	the USA	BNT162b2 or mRNA-1273	Patients with Inflammatory Bowel Disease	246
26	Connolly et al. [39]	Cross-sectional study	the USA	BNT162b2 or mRNA-1273	Patients with rheumatic and musculoskeletal diseases	325
27	Shimabukuro et al. [40]	Surveillance	the USA	BNT162b2	General population	9,943,247
				mRNA-1273	General population	7,581,429
28	Monin et al. [41]	Cohort Study	the UK	BNT162b2	Patients with cancer	140
				BNT162b2	Healthy population	40
29	Geisen et al. [42]	Cohort Study	Germany	BNT162b2 or mRNA-1273	patients with chronic inflammatory conditions	26
30	Polack et al. [43]	RCT	Argentina, Brazil, South Africa, etc.	BNT162b2	General population	21,621
31	Baden et al. [44]	RCT	the USA	mRNA-1273	General population	15,181
32	Cohen et al. [45]	Cohort Study	Israel	BNT162b2	Patients with cancer	728
33	El-Shitany et al. [46]	Cross-sectional study	Saudi Arabia	BNT162b2	General population	237
34	Sørvoll et al. [47]	Cohort Study	Norway	ChAdOx1	HCW	492
35	Cohen et al. [48]	Cohort Study	Israel	BNT162b2	Patients with cancer	137
36	Pottegård et al. [49]	Surveillance	Denmark, Norway	ChAdOx1	General population	281,264
37	China CDC, National Center for Adverse Drug Event Monitoring, China [50]	Surveillance	China	Chinese SARS-CoV-2 vaccine	General population	265,000,000
38	Hatmal et al. [51]	Cross-sectional study	Jordan	BIBP	General population	845
				BNT162b2	General population	605
				ChAdOx1	General population	686
39	Gras-Champel et al. [52]	Surveillance	France	ChAdOx1	HCW and patients	3,263,188
40	Simpson et al. [53]	Surveillance	the UK	ChAdOx1 or BNT162b2	General population	2,529,014
41	Huh et al. [54]	Surveillance	Korea	ChAdOx1	General population	8,548,231
42	Bikdeli et al. [55]	Surveillance	the UK, the USA	ChAdOx1 or Ad26.COV2.S	General population	21,200,000
43	Cari et al. [56]	Surveillance	Belgium, Denmark, etc.	ChAdOx1 or BNT162b2	General population	43,032,170
44	McMurry et al. [57]	Surveillance	the USA	BNT162b2 or mRNA-1273	General population	68,250
45	Julia et al. [58]	Surveillance	the UK	ChAdOx1	General population	19,608,008
				BNT162b2	General population	9,513,625
46	Cugno et al. [59]	Cohort Study	Italy	mRNA vaccine	HCW	3586
47	Baldolli et al. [60]	Cross-sectional study	France	BNT162b2	General population	2048
48	Chevallier et al. [61]	Cross-sectional study	France	BNT162b2	Allogeneic hematopoietic stem-cells recipients	94
				BNT162b2	Healthy population	24
49	Furer et al. [62]	Cross-sectional study	Israel	BNT162b2	patients with autoimmune inflammatory rheumatic diseases	673
				BNT162b2	Healthy population	121
50	Quiroga et al. [63]	Cross-sectional study	Spain	BNT162b2	HCW	565
				mRNA-1273	HCW	42
51	Riad et al. [64]	Cross-sectional study	Germany, Czech republic	ChAdOx1	HCW	92
52	Riad et al. [65]	Cross-sectional study	Turkey	CoronaVac	HCW	779
53	Vallée et al. [66]	Cross-sectional study	France	ChAdOx1	HCW	451

Notes: RCT stands for randomized controlled trial; BIBP stands for inactivated SARS-CoV-2 vaccine developed by Beijing Biologics Research Institute; WIBP stands for inactivated SARS-CoV-2 vaccine developed by Wuhan Biologics Research Institute; HCW stands for healthcare workers.

**Table 2 vaccines-10-00596-t002:** Meta-analysis results of incidence of adverse events (AEs) in observational studies.

Symptoms	Viral Vector Vaccine				mRNA Vaccine				Inactivated Vaccine			
	Number of Records	*I*^2^ (%)	*p*	Incidence % (95%CI)	Number of Records	*I*^2^ (%)	*p*	Incidence % (95%CI)	Number of Records	*I*^2^ (%)	*p*	Incidence % (95%CI)
AEs	4	95.3	<0.001	94.23 (90.84, 97.61)	15	100	<0.001	65.74 (52.13, 79.34)	2	93.4	<0.001	20.05 (10.91, 29.19)
SAEs	1	-	-	0.07 (−0.07, 0.21)	1	-	-	1.25 (0.52, 1.99)	2	0	0.982	0.07 (0.00, 0.13)
Death	0	-	-	-	6	17.8	0.298	0.03 (−0.02, 0.07)	0	-	-	-
Allergic symptoms												
Tongue edema	1	-	-	18.04 (17.03, 19.04)	1	-	-	3.61 (1.41, 5.81)	0	-	-	-
Angioedema	2	99.9	<0.001	9.58 (−9.13, 28.28)	2	88.4	0.003	2.24 (−1.52, 6.00)	0	-	-	-
Body bruises	1	-	-	4.96 (3.33, 6.58)	1	-	-	1.98 (0.87, 3.09)	1	-	-	2.37 (1.34, 3.39)
Urticaria	3	99.3	<0.001	2.99 (−0.76, 6.73)	7	73.9	0.001	0.91 (0.40, 1.41)	0	-	-	-
Anaphylaxis	1	-	-	0.95 (0.56, 1.33)	5	0	0.674	0.24 (−0.07, 0.55)	0	-	-	-
Cardiovascular and cerebrovascular symptoms												
Palpitation	1	-	-	28.29 (27.11, 29.47)	5	94.4	<0.001	2.99 (1.37, 4.61)	1	100	<0.001	0.09 (0.00, 0.18)
Irregular heartbeat	1	-	-	23.03 (19.88, 26.18)	1	-	-	11.90 (9.32, 14.48)	1	100	<0.001	7.81 (6.00, 9.62)
Abnormal blood pressure	1	-	-	13.85 (11.26, 16.43)	4	94.5	<0.001	2.47 (0.66, 4.29)	1	100	<0.001	5.44 (3.91, 6.97)
Chest discomfort	2	98.7	<0.001	13.21 (−0.46, 26.87)	6	93.9	<0.001	1.68 (0.69, 2.67)	1	100	<0.001	7.10 (5.37, 8.83)

## Data Availability

Not applicable.

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
