# Peer review of "Safety of Global SARS-CoV-2 Vaccines, a Meta-Analysis"

_vaccines, 2022, doi:10.3390/vaccines10040596_

Round 1

Reviewer 1 Report

This is brave work assessing the adverse events (AEs) of COVID-19 vaccine using a combination of systematic review and meta-analysis. The authors also compare the prevalence of various AEs among different vaccine platforms as well as ages and different study types (observation, RCTs etc). The authors report higher incidence of AEs in people aged <55 years old and certain AEs higher in mRNA platforms. These are interesting. However, I have some concerns.

MAJOR

  1. My major concern is with the inclusion criteria. Or instance the AEs in RCT of general population is pooled with those from observational and surveillance studies targeting specific populations such as patients that underwent organ transplantation, health care workers and patients with specific diseases.

In this type of systematic review,one would expect ficus to be on RCTs where vaccinated are compared with volunteers that received placebo to understand the dynamics of AEs because studies of focused groups will naturally introduce bias in the findings (comfirned by the heterogeneity which is close to 100% in some cases, publication bias via Egger’s test, and captured by the sensitivity test).   

Indeed, some of the patients (especially transplant patients) may have compromised immune systems while HCW may suffer due to re-infection etc.

  1. Another concern with this study is the inclusion of observational and survellance studies which in the context of the aim of the systematic review may be biased in effective comparison of the adverse events such that one cannot expect a adverse event in vaccinated people since they were not injected with anything. This is very evident in Lines 183-193 whereby the authors compare adverse event between vaccinated and unvaccinated people.

For RCT (lines 195 - 206), the argument is clear that people who received placebo showed less adverse event indeed, but comparison with unvaccinated is just misleading.

  1. Please look again at the title. “Vaccine should be “vaccines” and novel coronavirus could be COVID-19 or SARS-CoV-2 for clarity.
  2. Authors need to provide definitions of adverse events and compare it with adverse effects. This will help to clarify whether the events reported may be driven or due to other factors and not necessarily due to the vaccines innoculation. This should be expanded in the discussion and alighted as part of the limitation.
  3. Literature quality evaluation: for each type of study, please describe the quality assessment scores to give a feel for how the studiy types perform and the implication on the high ehetrogeneity later deduced?
  4. The manuscript would benefit from an extensive English language proofing (e.g., Line 13, 15, 92, 94, 95, 103 etc).
  5. The discussion is simply a repetition of the result and does not do well to put the findings in the context of previous works and findings.
  6. Finally, high heterogeneity is sometimes a sign that a meta-analysis is not feasible and performing it may be misleading. One way to resolve this is by limiting the study to a systematic review alone without meta-analysis. However, this is merely my opinion.

MINOR

  1. [email protected] is not an affiliation
  2. Line 12-13, spell out COVID-19 first and then abbreviate, not the other way round.
  3. Line 99: Please specify the initials of the authors that did what.
  4. Lines 101-102: please clarify.
  5. Lines 152 – 153: the percentages of studies from at least tpw of the main regions should be provided in parenthesis.
  6. Lines 165-166: please clarify the statement "This paper integrated the results of multiple articles that the re-165 search methods, sample size and sensitivity were different, and the total incidence of AEs showed great heterogeneity. "
  7. Line 348: do you mean "strong" and not "stung"?

Author Response

Thank you for your valuable advice. We considered your suggestions carefully and revised our manuscript. We focus on fewer AEs, discuss the results in the context of previous works and findings, and discuss the limitation in depth. We hope this will help to interpret the findings of our study. We also revised corresponding tables and figures. The detailed reply is in the attachment. We look forward to your comments.

Reviewer 2 Report

English is poor - needs to be extensively edited

Was there a protocol written for the study? Was it published?

Need to provide search code for each database searched in an appendix

Method for choosing random or fixed effects analysis flawed

Suggest excluding cross-sectional studies as I am not sure how AEs can be properly assessed without follow up?

RCT risk of bias assessment is not numerical and should not be summed

Too much heterogeneity to allow meta-analysis

Should show forest plots

Too much information presented - suggest focusing on deaths, SAE and "abnormal" events

I found the findings to be uninterpretable due to all the issues outlined above

Author Response

Thank you for your valuable advice. We considered your suggestions carefully and revised our manuscript. We focus on fewer AEs, discuss the results in the context of previous works and findings, and discuss the limitation in depth. We hope this will help to interpret the findings of our study. We also revised corresponding tables and figures. The detailed reply is in the attachment. We look forward to your comments.

Response to Reviewer 2 Comments

Point 1: English is poor - needs to be extensively edited.

Response 1: Thank you for your advice. In the revised manuscript, the language was edited by a native English speaker from beginning to end..

Point 2: Was there a protocol written for the study? Was it published?

Response 2: Thank you for your question. There is no protocol written.

Point 3: Need to provide search code for each database searched in an appendix.

Response 3: Thank you for your suggestion. We have added the search code in appendix in line 498-550.

Point 4: Method for choosing random or fixed effects analysis flawed

Response 4: Thank you for your advice. We carefully studied the "Cochrane Handbook for Systematic Reviews of Interventions", section 10.10.4, "Incorporating heterogeneity into random-effects". It mentioned that many have argued that the decision should be based on an expectation of whether the intervention effects are truly identical, preferring the fixed-effect model if this is likely and a random-effects model if this is unlikely (Borenstein et al 2010). Since it is generally considered to be implausible that intervention effects across studies are identical (unless the intervention has no effect at all), this leads many to advocate use of the random-effects model. Therefore, we changed our method to using random-effects for all AEs meta-analysis (line 139).

Point 5: Suggest excluding cross-sectional studies as I am not sure how AEs can be properly assessed without follow up?

Response 5: Thank you for your advice. In cross-sectional studies, vaccinated population was asked to complete questionnaire focusing diagnoses and reactions after they receive vaccines. Surveillance study collected total administered doses and AEs reported by vaccine receivers in a time period to calculate incidence of AEs. Although researchers only collected response once, there exist time series between vaccination and AEs. It is kind of like retrospective cohort study without comparison with control groups.

Point 6: RCT risk of bias assessment is not numerical and should not be summed.

Response 6: Thank you for your suggestion. We have revised the assessment. We described the detailed items that was not met by any RCTs and made an overall evaluation of risk of bias. (line 168-174). All RCTs met the requirements of blinding of participants and personnel, and selective reporting. Study 11 and 30 did not clarify the method used to generate the allocation sequence. Study 30 did not describe the method used to conceal the allocation. Study 4 did not describe the completeness of outcome data. Five RCTs did not mention how outcome assessors were blinded from knowledge of which intervention a participant received. We also supplement risk of bias graph (Figure 2).

Point 7: Too much heterogeneity to allow meta-analysis.

Response 7: [Reply] Thank you for your advice. We admit there was too much heterogeneity. We used random-effect model in order to obtain reliable results under the high heterogeneity and helped to understand the incidence rate of AEs. We stuck to strict inclusion criteria and carried out quality assessment to control heterogeneity. We also carried out sensitivity analysis and the results were robust except few AEs. We also added the discussion of high heterogeneity in "limitation".

Point 8: Should show forest plots.

Response 8: Thank you for your valuable advice. We added forest plot of AEs in observational studies in article (Figure 3).

Point 9: Too much information presented - suggest focusing on deaths, SAE and "abnormal" events.

Response 9: Thank you for your valuable advice. We agree that it would be better to focus on those abnormal events. We have revised our article to only reporting AEs, SAEs, death, allergic symptoms and cardiovascular disease. We also added the reason why should those AEs be the focus of our study in "background" (line 83-85) and revised the discussion.

Point 10: I found the findings to be uninterpretable due to all the issues outlined above.

Response 10: Thank you for your valuable advice. We have revised our article by focusing on fewer AEs, discussing the results in the context of previous works and findings, and discussing the limitation in depth. We hope this will help to interpret the findings of our study from beginning to end.

Reviewer 3 Report

The study from Chen et al., describes in a very organized way, a meta-analysis study for different Coronavirus vaccines. This is a nice work as it provides information regarding the different vaccines used for COVID-19.

An introductory paragraph explaining in a bit more detail the differences of the difference type of vaccines (or maybe a chart) might be useful for the study.

Author Response

Thank you for your valuable advice. We considered your suggestions carefully and revised our manuscript. We focus on fewer AEs, discuss the results in the context of previous works and findings, and discuss the limitation in depth. We hope this will help to interpret the findings of our study. We also revised corresponding tables and figures. The detailed reply is in the attachment. We look forward to your comments.

Response to Reviewer 3 Comments

Point 1: The study from Chen et al., describes in a very organized way, a meta-analysis study for different Coronavirus vaccines. This is a nice work as it provides information regarding the different vaccines used for COVID-19.

An introductory paragraph explaining in a bit more detail the differences of the difference type of vaccines (or maybe a chart) might be useful for the study..

Response 1: Thank you for your advice. We added a paragraph introducing the technique of the five vaccine develop platform in line 59-69. In general, mRNA vaccines transport the mRNA that encodes virus antigen into host cell; inactivated vaccines were made by killed viruses and was with good safety; viral vector vaccines use weak or non-pathogenic viruses as vector and integrate antigen genes of target virus into the genome of the vector.

Round 2

Reviewer 2 Report

Thank you for addressing most of my previous comments / suggestions. However the English is still not acceptable. And given the very high heterogeneity, meta-analysis does not seem appropriate. I still find the results uninterpretable. Why is the vaccine safety acceptable? Please define acceptable. Funnel plots should not be used when there is <10 observations. 

Author Response

Thank you for your valuable advice. We edited the language again and studied the method of dealing with heterogeneity. We replied your concerns in a Word, please see the attachment. Thank you very much!

Response to Reviewer 2 Comments

Point 1: Thank you for addressing most of my previous comments / suggestions. However the English is still not acceptable. 

Response 1: Thank you for your advice. We invited another native English speaker to edit our manuscript and modified our language carefully.

Point 2: And given the very high heterogeneity, meta-analysis does not seem appropriate. I still find the results uninterpretable. Why is the vaccine safety acceptable? Please define acceptable.

Response 2: Thank you for your suggestion. We admitted the problem of high heterogeneity in our meta-analysis could impair reliability. In the case of high I2 , we applied a random-effects model. The high I2 is probably attributable to differences between studies (e.g. differences in study design, patient populations, diagnostic methods, application of interventions, or definitions of outcome). The high I2, however, will have broad CIs if a random-effects method is used, which will contain the true effect in approximately 95% of the simulations. By using random-effects model, we could confirm what has been already known (pooled resultes of published studies) but could not predict the result of next large study[1]. Therefore, we think that our results is meaningful in showing the level of vaccine safety to a certain extent. We also conducted meta-regression analysis including the factors of study types, vaccine types, populations, age, sex, doses, and the results showed that study types impacted the incidence of AEs significantly. We have used subgroup analysis to control the heterogeneity as far as possible. Given the heterogeneity in the study, we discussed this limitation in discusion and interpreted the result cautiously. Our result also indicated the importance of focusing the comparability in future studies. We also found a meta-analysis of percentage of asymptomatic SARS-CoV-2 infections which was published on JAMA Network Open, whose I2 was 99%[2].

Ref:

[1] Melsen WG, Bootsma MC, Rovers MM, Bonten MJ. The effects of clinical and statistical heterogeneity on the predictive values of results from meta-analyses. Clin Microbiol Infect. 2014 Feb;20(2):123-9. doi: 10.1111/1469-0691.12494. PMID: 24320992.

[2] Ma Q, Liu J, Liu Q, Kang L, Liu R, Jing W, Wu Y, Liu M. Global Percentage of Asymptomatic SARS-CoV-2 Infections Among the Tested Population and Individuals With Confirmed COVID-19 Diagnosis: A Systematic Review and Meta-analysis. JAMA Netw Open. 2021 Dec 1;4(12):e2137257. doi: 10.1001/jamanetworkopen.2021.37257. PMID: 34905008; PMCID: PMC8672238.

We concluded that the safety of vaccines was acceptable because most AEs were mild and transient, and the incidence of SAEs was low (line 343). Eventhough some vaccines may have higher risk in vascular diseases, the benefit outweigh their safety risks since they passed the examination of WHO’s Strategic Advisory Group of Experts on Immunization (SAGE), which formulates vaccine specific policies and recommendations for vaccines’ use in populations.

Point 3: Funnel plots should not be used when there is <10 observations.

Response 3: Thank you for your suggestion. We have deleted the funnel plots of <10 observations.